# Plant Growth Promoting Rhizobacteria (PGPR) and Arbuscular Mycorrhizal Fungi Combined Application Reveals Enhanced Soil Fertility and Rice Production

Delai Chen [1,2,*], Munawar Saeed [3], Mian Noor Hussain Asghar Ali [4], Muhammad Raheel [5], Waqas Ashraf [5], Zeshan Hassan [6], Muhammad Zeeshan Hassan [7], Umar Farooq [7], Muhammad Fahad Hakim [7], Muhammad Junaid Rao [8], Syed Atif Hasan Naqvi [7,*], Mahmoud Moustafa [9,10], Mohammed Al-Shehri [9] and Sally Negm [11,12]

1  College of Life Science and Technology, Longdong University, Qingyang 745000, China
2  Gansu Key Laboratory of Protection and Utilization for Biological Resources and Ecological Restoration, Qingyang 745000, China
3  Department of Soil Science, Faculty of Agricultural Sciences and Technology, Bahauddin Zakariya University, Multan 60800, Pakistan
4  Department of Farm Structures, Faculty of Agricultural Engineering, Sindh Agriculture University, Tandojam 70060, Pakistan
5  Department of Plant Pathology, Faculty of Agriculture and Environment, The Islamia University of Bahawalpur, Bahawalpur 63100, Pakistan
6  College of Agriculture, Bahauddin Zakariya University, Multan Bahadur Sub Campus Layyah, Multan 31200, Pakistan
7  Department of Plant Pathology, Faculty of Agricultural Sciences and Technology, Bahauddin Zakariya University, Multan 60800, Pakistan
8  State Key Laboratory for Conservation and Utilization of Subtropical Agro-Bioresources, Guangxi Key Laboratory of Sugarcane Biology, College of Agriculture, Guangxi University, Nanning 530001, China
9  Department of Biology, Faculty of Science, King Khalid University, Abha 61413, Saudi Arabia
10 Department of Botany and Microbiology, Faculty of Science, South Valley University, Qena 83511, Egypt
11 Department of Life Sciences, College of Science and Art MahyelAseer, King Khalid University, Abha 61413, Saudi Arabia
12 Unit of Food Bacteriology, Central Laboratory of Food Hygiene, Ministry of Health, Branch in Zagazig, Sharkia 44511, Egypt
*  Correspondence: cdl829@126.com (D.C.); atifnaqvi@bzu.edu.pk (S.A.H.N.)

**Abstract:** Rice (*Oryza sativa* L.) is an important crop that is grown worldwide to supply the world's expanding food demand. In the current study, the effects of plant growth-promoting rhizobacteria (PGPR) and *Arbuscular mycorrhizal* fungi (AMF) on soil fertility and rice growth were explored. Rice plants were inoculated to evaluate how AMF fungi and PGPR affect various aspects of soil and plants, implicating abiotic stress tolerances. The experiment was carried out in a completely randomized design with three replicates under the controlled conditions. Results depicted that the plants that were inoculated with a mixture of AMF and PGPR had better yields and nutritional concentrations, while both AMF and PGPR lowered soil pH and organic matter differently. Similarly, AMF and PGPR treatments significantly increased the amount of N, P, K, and B in the post-harvest soil. The PGPR-inoculated plants had a 10–40% higher buildup of N in their tissues. Similarly, when they were compared with non-infected plants, AMF-inoculated treatments demonstrated a greater N accumulation in the rice tissue. The maximum P content in plant tissues was 0.149% in PGPR5-infected plants, either alone or in combination with AMF. In T12, AMF + PGPR5 inoculated rice plants, the maximum K uptake was 1.98%, which was 54% higher than the control treatment. The sole application of AMF raised K buildup in rice tissues by 38% compared with the control treatment. The improved productivity of plants with AMF and PGPR (especially with PGPR5) was attributed to the increased availability of nutrients in the soil. As a result, rice plant growth, yield, and essential element uptakes were boosted significantly. The present study's results suggested using the combined application of AMF + PGPR5 for improving the rice yield and for sustaining the soil health.

**Keywords:** rice; *Arbuscular mycorrhizal* fungi; PGPR; nutrients; yield; growth attributes

## 1. Introduction

Rice is an edible cereal grain and a grassy plant that belongs to the Poaceae family. The contribution of rice to Pakistan's agricultural sector is 3.1%, while for the country's GDP it is responsible for 0.6% [1]. As reported by the FAO, the food balance sheet of 1996 reported that in Indonesia, Bangladesh, and Vietnam rice provided about sixty percent of the calories, while it provides fifty to sixty percent of calories in China, Korea, and Thailand. In the case of Pakistan and America, it provides twenty to thirty percent of calories [2,3]. Different traits have been attributed to rice, including form, production, weight, and biomass [4]. Affirmation about the initial production of rice grains in China and other Southeast Asian countries has been found, but primitive evidence from about 7000–5000 years BC confirms China as the very first cultivator of the rice crop. Asian countries such as Pakistan, China, Bangladesh, and Indonesia provide about 90% of the rice, along with other Asian countries contributing a little bit, such as Japan [2–5]. Due to the presence of vitamins, proteins, and other nutrients such as Ca, P, and Fe, rice is thought to be a healthy food.

Plant growth-promoting rhizobacteria (PGPR) are bacteria that colonize the rhizosphere and enhance the growth of plants through different actions [6,7]. Moreover, PGPR play a direct role in nitrogen-use efficiency, plant hormone synthesis, the enhanced solubility of minerals in the soil such as P, and the formation of siderophores that chelate iron and make it available to the rhizosphere of plants, in addition to the improved growth of the plant [8–10]. The ability of PGPR in solubilizing the inorganic and organic P in soil has been studied and discussed [11,12]. Because of the ability of PGPR to colonize more than a hundred plant species, it remarkably improves the growth, production, and yield of rice and many other crops [13,14]. The *Azospirillum* genus is the most commonly used commercial biofertilizer and one of the most important genera [15].

On the other hand, a well-known symbiotic association on the earth is found between plants and the *Arbuscular mycorrhizal* fungi (AMF) [16], which has proven to be beneficial for the plants as it provides plants with a nutrient-rich environment, mainly consisting of phosphate [17,18], while the other partner plant provides the carbohydrates to the symbiotic relationship partners, i.e., AMF, which they synthesize during photosynthesis [19,20]. In addition, some species of rice called transgenic species contain genes to resist the pests' release of some secretions into the soil microbiome that contains a special protein which is exclusively related to pesticides [21,22]. In return, AMF contain a special mycelium that permits the transfer of C-products during photosynthesis to the soil microbiome. The growth of bacteria can be enhanced in the soil by the mycelium and living hyphal exudates that are carbohydrates because of their higher turnover rate [23,24]. Meanwhile, the growth of the plant is enhanced due to its beneficial association with AMF. Thus, due to this increase in growth, the hyphae present in the soil absorb the maximum amount of nutrients [18–25]. Interconnected hyphae around the roots of a plant are reported to harm the roots of the nearby plant [26,27]. Mycorrhizal association is a symbiotic relationship between plants and fungi, where the fungi provide nutrients and water to the plant in exchange for sugars produced through photosynthesis. This relationship affects the growth rate of plant roots as it improves their ability to absorb nutrients and water, leading to increased root growth. On the other hand, xylem pressure, which refers to the pressure of water in the plant's xylem tissue, does not have a direct impact on the growth rate of roots [28]. Species of AM fungi can enrich the soil with nutrients, resulting in a good production of plants by direct or indirect involvement [29,30]. Therefore, to enhance the effect on soil texture, soil structure, and nutrients, a good relation of prokaryote (bacteria) composition and growth responses (when a new variety of rice is being grown where AMF association is found in soil) is needed.

Additionally, inoculating plants with AMF improved plant water and nutrient uptake [31]. Most importantly, through their hyphae (that have enormous surface areas on which the extra radical hyphae function as a bridge between the soil and the plant roots), AMFs have the capacity to scavenge available P [32,33]. Using AMF can improve plant growth and yield. Organosulfides, polyphenols, phytosterols, stilbenes, vitamins, lignin, and terpenoids, including carotenoids, are only a few examples of pharmaceutical chemicals whose productivity has improved and could be beneficial to human health due to their antioxidant properties [16,33]. Additional benefits provided by the AMF to plants include improved enzymatic production, increased photosynthesis, and improved soil fertility through symbiotic or associative nitrogen fixation by bacteria [34,35].

Hence, the interaction between AMF and PGRP is one of the primary functions of these crucial plant-associated symbionts. It is a well-known fact that soil bacteria in the mycorhizosphere use AMF hyphae as a source of energy for the production of C products [36]. To boost the rate and extent of root plant colonization by AMF, PGPR can promote the germination of AMF spores. In wheat, rice, and black gram, the combined application of PGPR and AMF promote the plant yield by up to 41% compared with non-inoculated plants. Plant growth parameters such as grain number, grain weight, plant height, and biomass production increased significantly with AMF and PGPR addition [15,32,35]. Different strains of PGPR have different modes of action. Therefore, the present study aimed to examine the combined and sole inoculation effects of PGRP and AMF on various indicators of rice growth, nutrient uptake, and soil fertility enhancement. Nature is the best source for any remedy or problem faced by agriculture. In order to deal with the soil problems and to improve the crop productivity, PGPR and AMF (found in the soil as natural microbiome that were collected from the local soil vicinities in the fields and from the plant rhizosphere) offered the best source of solution for the domestic soils for both soil problems and yield improvement. In PGPR, there is a great diversity due to the presence of sexual reproduction among their populations through conjugation, transformation, and transduction and they can play very positive roles for soil health. Hence, experiments were carried out with the locally obtained strains of PGPR and AMF to observe their efficacy for the domestic issue of the soil and the productivity of the rice crop.

## 2. Materials and Methods

### 2.1. Preparation and Application of AMF and PGPR Inoculum

The rice nursery was grown at the agriculture experimental farm, Bahauddin Zakariya University, Multan, Pakistan, with 10 kg of sandy loam soil filled in earthen pots (10″ ×45″). For the AMF inoculation, approximately 5-cm-deep holes were made in the experimental pots. Then, approximately 10 g of mycorrhizal inoculum predominantly containing the *Glomus* species along with 9 propagules (including the spores, hyphal fragments, and root portions) of *Gigaspora albida* (Clonex® Root Maximizer; Bustan, ON, Canada) as inoculum was used in this study as it was predominantly rich in the *Glomus* species and was applied in those 5-cm-deep holes. In the AMF control soil, thiophanate-methyl (70% WP) was applied at 50%. After applying the AMF treatment, the rice seedlings (dipped in the sugar solution) were transferred to the pots. The application of AMF was also made at the edges of the earthen pots to which the plants were transferred so that the AMF inoculum could reach the dispensing roots. On the other hand, the application of plant growth-promoting rhizobacteria (PGPR) viz., *Paenibacillus* (PGPR 1), *Rhizobium* (PGPR 2), *Bacillus* (PGPR 3), *Azotobacter* (PGPR 4), and *Pseudomonas* (PGPR 5) were used to carry out the standard process. Briefly, the solution of the sugar and water (0.25:1 ratio) was prepared. After that, in the solution of sugar and water, 80 mg of PGPR inoculum containing $10^6$ cfu/mL calibrated at spectrophotometer of each was added in the current research. The roots of the plants were dipped into the solution, from where the bacterial strains were connected to the roots and then transplantation into the pots was conducted [37]. All treatments were arranged in triplicates.

## 2.2. Soil Characterization

Soil sampling was performed to check the chemical and nutritional composition and the health of the soil, viz., soil pH, EC, and organic matter, before filling the pots. For pH and EC determination, the mixture of distilled water (DW) and soil (1:1) was mixed comprehensively, was kept overnight, and then pH was noted through a pH meter. In addition, the EC meter was standardized using 0.01 N KCl solution and EC was noted by using the probe of the EC meter.

A titration method was used for this purpose; 10 mL of 1 N potassium dichromate was added to 1 g of soil. It was cooled down with 30 min breaks, with the addition of 20 mL of pure sulfuric acid. A total of 10 mL of concentrated ortho-phosphoric acid and 200 mL of distilled water were added to the solution and cooled down again. After cooling the samples, 10–15 drops of diphenylamine were added which adjusted the concentration of 0.5 M ferrous ammonium sulfate until the violet–blue tone turned green as described by Jenkinson [38,39]. The following formulas were used to compute the percentage of organic matter:

$$M = \frac{10}{V_{blank}}$$

$$\text{Oxidizable Organic Carbon (\%)} = \left[V_{blank} - V_{sample}\right] \times 0.3 \times \frac{M}{Wt}$$

$$\text{Total Organic Carbon (\%)} = 1.334 \times \text{Oxidizable Organic Carbon (\%)}$$

$$\text{Organic Matter (\%)} = 1.724 \times \text{Total Organic Carbon(\%)}$$

where M is the ferrous ammonium sulfate molarity; Wt. is the air-dried soil weight; $V_{sample}$ is the ferrous ammonium sulfate solution utilized for titration of the sample; $V_{blank}$ is the ferrous ammonium sulfate solution utilized to titrate blank; 0.3 is the carbon equivalent weight.

## 2.3. Soil Nitrogen (N), Extractable Phosphorus (P), Potassium (K), and Boron (B)

For the determination of soil nitrogen (N), 05 g of soil and 1 g of the combination of catalysts were placed into the digestion tube, followed by overnight incubation according to the hood method [40]. Briefly, 10 mL of sulfuric acid was added and digestion tubes were placed in the digestion unit until the colorless endpoint liquid was obtained. The solution was distilled by a distillation unit. Following that, 20 mL of 4% boric acid was applied to a beaker to collect the distillate. The indicator was then added in groups of three or four drops. At first, the solution was blue, then it turned purple, and last, it turned golden yellow. The golden solution was titrated against 0.1 N sulfuric acid until the terminus of the solution changed to purple. N was calculated by using the following formula:

$$N(\%) = \frac{14.1 \times \text{Vol. of sulphuric acid used for sample} - \text{Vol. of sulphuric acid used for blank} \times \text{N of acid}}{\text{Wt. of sample (g)} \times 10}$$

The soil extractable phosphorus was determined according to the method used by Sharpley [37]. Briefly, two reagents were used for this purpose. For the preparation of reagent A, 250 mL of distilled water was used to liquefy the 12 g of ammonium hepta-molybdate. A total of 0.2908 g of potassium antimony titrate was dissolved in 100 mL of distilled water in another conical flask. Both of these liquids were then well-combined. The whole solution was then increased to 2 L by taking 1 L of 5 N sulfuric acid and adding it to this solution. Then, 200 mL of reagent A was added with 1.056 g of L-ascorbic acid to create reagent B. Following the addition of 1 g of dirt, 20 mL of a separating solution containing 0.5 M sodium bicarbonate and 30 min of mechanical shaking were added. Then, 2 mL of extract and 1.6 mL of reagent B and 6.4 mL of distilled $H_2O$ were mixed in the whirlpool mixture. The whole volume of the solution was 10 mL. The series of standard solutions of 1,2,3,4, and 5% concentrations were prepared using potassium dihydrogen phosphate and were conducted using a spectrophotometer at an 882 nm wavelength. The value of $R^2$ that was almost equal to 1 was estimated. Following the measurement, the prepared 10 mL

solution's reading was obtained by running the sample through a spectrophotometer set to wavelength 882 nm. The calculation was completed at the conclusion to determine the amount of accessible phosphorus:

$$Extractable\ P(ppm) = ppm\ P\ (from\ calibration\ curve) \times \frac{V}{Wt} \times \frac{V_2}{V_1}$$

where $V$ is the volume of the soil extract, $V_1$ is the volume of the used soil extract, $V_2$ is the volume of the used flask, and $W_t$ is the air-dried soil weight.

For the determination of potassium, the reagent used in the experiment was prepared according to the method described by Beckett [39], with 1 N ammonium acetate solution and a standard solution of 1 N potassium chloride. Five grams of air-dried soil was added into 25 mL of 1 N ammonium acetate solution for the extraction process. It was left on the mechanical shaker for 30 min at 200–300 rpm. The filter paper was used to filter the suspension (Whatman No. 1). A flame photometer was used to test a series of standard solutions with concentrations of 20, 40, 60, 80, 100, 150, and 200%. A graph was created with an $R^2$ value that was almost equal to 1. Reading was observed and the following calculation was used to determine the extractable K:

$$Extractable\ K(\%) = \%\ K\ (from\ calibration\ curve) \times \frac{V}{Wt}$$

where $Wt$ is the air-dried soil weight and $V$ is the soil extract volume.

For the determination of boron, boron-free, 0.2 g of activated charcoal was added to10 g of 2 mm sieved soil. The flask spent five minutes on the hot plate after the addition of 20 mL of distilled water for the boiling purpose. The suspension was put through a filter made of Whatman No. 1 filter paper. Then, an aliquot of 1 mL, 2 mL of the buffer solution, and 2 mL of the azomethine-H solution were put into a 10 mL polypropylene tube. After 30 min, readings of the spectrophotometer at the 420 nm wavelength for the blank, the standard, and the samples were recorded [41]. The calculation of boron used the following formula:

$$B(\%) = \%\ B\ from\ calibration\ curve \times \frac{V}{Wt}$$

### 2.4. Experimental Design of Pot Trial

Rice seedlings were grown in sandy loam soil. Each pot of (10″ × 45″) size was filled with 10 kg of soil that was previously passed through a 2 mm mesh sieve. In the current experiment, 5 strains of PGPR were examined, while 1 level of AMF was used to evaluate their effects on rice yield and soil fertility. The treatments were as follows: T1 = control; T2 = PGPR1 (RB1); T3 = PGPR2 (RB2); T4 = PGPR3 (RB3); T5 = PGPR4 (RB4); T6 = PGPR5 (RB5); T7 = AMF; T8 = PGPR1 (RB1) + AMF; T9 = PGPR2 (RB2) + AMF; T10 = PGPR3 (RB3) + AMF; T11 = PGPR4 (RB4) + AMF; T12 = PGPR5 (RB5) + AMF. All of the treatments were applied in a completely randomized design (CRD) following three replicates. Moreover, recommended doses of fertilizers were added at a rate of N100: P70: K50 per hectare. Considering the suggested dosages, 1.38 g of SSP, 0.92 g of Urea, and 0.79 g of SOP were added to each pot.

### 2.5. Plant Leaf Analysis

Plant leaf samples were prepared by first drying them out and then powdering them. For digestion purposes, the di-acid (mixture of nitric acid and perchloric acid) was mixed with a ratio of 2:1, and 10 mL di-acid was added in a 1 g plant sample. It was kept overnight for pre-digestion. For digestive reasons, a flask was placed on a hot plate and the temperature was gradually raised to 200 °C until thick vapors began to emerge. The solution's volume was then increased to 50 mL by the addition of distilled water. The solution was left behind to analyze various nutrients, viz., leaf nitrogen, phosphorus, potassium, and boron, in the leaf tissues.

For plant nitrogen, 1 g of prepared plant material and 1 g of catalyst mixture were added to the digestion tube and kept overnight. The next day, after adding 10 mL of sulfuric acid, tubes were placed on the digestion unit and samples were heated continuously until the colorless liquid was obtained, as described by Emmett [42]. Following that procedure, the solution was distilled using the distillation equipment. The distillate was collected in the beaker using 20 mL of 4% boric acid. Then, 3–4 drops of the indicator were added to the distillate and, as a result, the distilled solution's color changed to purple before being transformed into golden yellow during the distillation process. The solution, which was a golden yellow color, was titrated against 0.1 N sulfuric acid. At that moment, the solution was titrated until the endpoint color of the solution was purplish.

$$N(\%) = \frac{14.1 \ \times \ Vol. \ of \ sulphuric \ acid \ used \ for \ sample - Vol. \ of \ sulphuric \ acid \ used \ for \ blank \times N \ of \ acid}{Wt \ of \ sample \ (g) \times 10}$$

For the phosphorus measurement, distilled water was added to the flask, along with 10 mL of the plant-digested solution and 10 mL of the reagent ammonium hepta-molybdate vanadate [43]. Ten milliliters of the ammonium vanadate reagent was added to a series of standards that had concentrations of 0.5, 1, 1.5, 2, and 2.5%. Standards were run on a spectrophotometer at a wavelength of 480 nm after 30 min. These readings were then used to create a graph; it was discovered that the $R^2$ value was almost equal to 1.Following the testing of the standard solutions, the prepared plant sample solutions were placed in the spectrophotometer, readings were collected, and the contents of the P were calculated using the formula below:

$$P(\%) = \% \ P(from \ calibration \ curve) \times \frac{V_1}{Wt} \times \frac{100}{V_2} \times \frac{1}{10000}$$

where $V_1$ is the total digested volume of plant material, $V_2$ is the digested plant volume used in the analysis, and $W_t$ is the weight of plant material used for digestion. For the purpose of determining the amount of potassium present in the plants, a 1 g sample of ground plant material was placed in a porcelain cup and placed in a muffle furnace at a temperature of 550–600 °C for 5–6 h. The plant sample's ash was then mixed with 5 mL of 2 N HCl for thorough digestion. Following this, the solution was filtered and the filtrate was then saved for K measurement. A flame photometer (PFP 7, Jenway) was used for K quantification.

For boron, according to William's approach, the dry ashing procedure was used to determine B from the plant sample [43,44]. In a porcelain crucible, 1 g of dry powdered material was used for this. The samples were then ignited at 550 °C by placing the crucible in a muffle furnace. After ashing, 10 mL of a 0.36 N sulfuric acid solution was added to the porcelain crucibles, followed by 5 drops of DI water to moisten the ash. The crucibles were then heated for 20 min in a steam bath. After an hour at room temperature, samples were agitated with a plastic rod to break up any ash clods. The samples were then put through a Whatman No. 1 filter paper. The volume was generated by placing 1 filter paper into the 50 mL poly-propylene flask. A pipette was used to transfer 1 mL of the extract aliquot into a 10 mL polypropylene tube. The tube was then filled with 2 mL of the buffer solution. A total of 2 mL of the azomethine-H solution was then added and thoroughly mixed. After the spectrophotometer had been turned on for 30 min, samples were run and measurements at a 420 nm wavelength were recorded after the standard sample and the blank sample had been used. The following formula was used to determine the B:

$$B(\%) = ppm \ B \ from \ calibration \ curve \times \frac{V}{Wt}$$

### 2.6. Agronomic Parameters

The agronomic parameters, such as plant height (cm), spike length (cm), fresh weight (g), dry weight (g), 1000-grain weight (g), root length, number of tillers, and number of spikes, were measured from each treatment at the maturity stage. The rice crop was harvested in November 2022 and the harvested biomass was stored in paper bags for further assessment.

## 2.7. Data Statistical Analysis, Pearson Correlation, and PCA

The presented data were analyzed through one-way ANOVA using the Origin 2021 Pro tool to measure the extent of significance among the tested treatments. The data showed an average of three repeats for each treatment. The level of significance was measured by adopting Tukey's HSD test at a 5% probability level. Pearson correlation and PCA were also determined to measure the relativity between and among the treatments.

## 3. Results

### 3.1. Soil pH, EC, and Organic Matter

Both the combined and the sole application of PGPR (RB) and AMF had a significant ($p < 0.05$) impact on the soil pH. AMF inoculation decreased the soil pH by up to one unit. The maximum reduction in the soil pH was observed in the AMF + PGPR1 (RB1)- and the AMF + PGPR3 (RB3)-inoculated strains, whereas the highest pH was recorded in the AMF + PGPR3 strain compared with the control. The sole application of PGPR1 (RB1) and PGPR4 (RB4) raised the soil pH in comparison with the control, while reducing the soil pH when applied with AMF. In the case of PGPR2 (RB2) and PGPR5 (RB5), the effect was non-significant among sole and combined applications of PGPR5 (RB5) with AMF. However, the sole application of PGPR3 (RB3) reduced the soil pH and, in combination with AMF, the pH of the soil increased (Figure 1A).The inoculation of sole and combined PGPR (RB) and AMF reduced the organic matter concentration in the soil. Statistical analysis of the soil showed that PGPR (RB) inoculation imparted more effect on the soil organic matter than AMF. Although AMF significantly reduced the OM contents more than the control treatments, their combined effect was more significant, whereas higher OM contents were observed in the control treatment. The application of microbes of either PGPR (RB) or AMF organic matter contents were reduced due to the rapid decomposition of OM by microbes. Minimum OM contents were reported in the AMF and PGPR1 (RB) strains, reflecting a higher OM decomposition rate in that treatment (Figure 1C; Table 1).

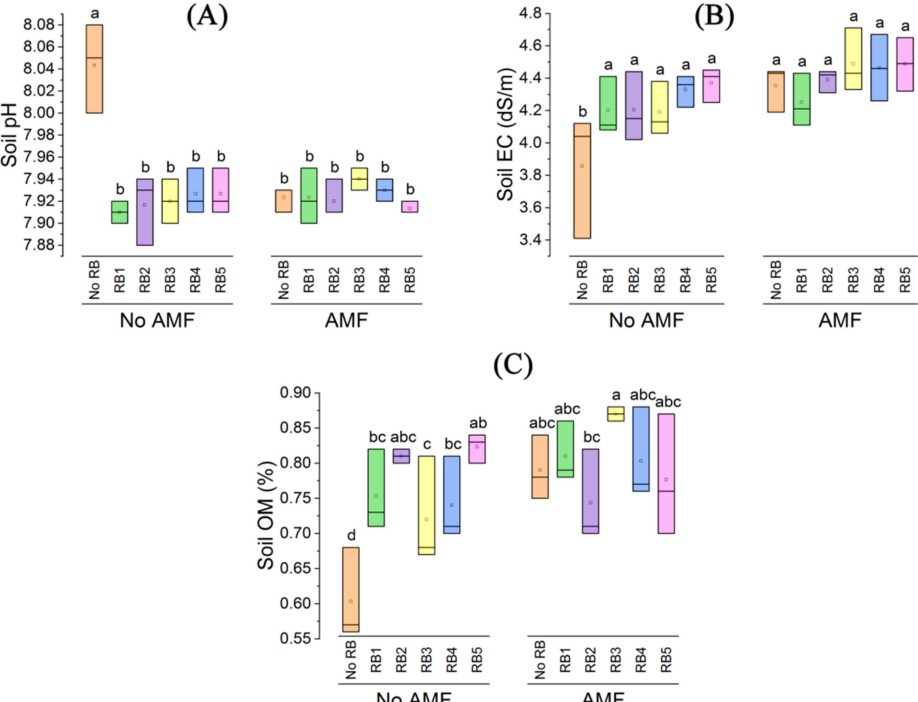

**Figure 1.** Effect of various strains of PGPR (RB1 = PGPR1, RB2 = PGPR2, RB3 = PGPR3, RB4 = PGPR4, RB5 = PGPR5) and AMF inoculation on soil pH (**A**), EC (**B**), and organic matter (**C**). The bars having different letters indicate significant differences from each other at $p < 0.05$.

**Table 1.** Experimental soil chemical characterization of various features.

| Parameters | Values |
|---|---|
| Soil pH | 8.2 |
| Soil EC (dS m$^{-1}$) | 1.1 |
| Soil organic matter (%) | 0.8 |
| Soil nitrogen (%) | 0.04 |
| Plant available phosphorous (%) | 6.5 |
| Extractable potassium (%) | 180 |
| Soil B (%) | 0.39 |

*3.2. Soil Analysis*

Nitrogen contents in the soil, viz., soil nitrogen, phosphorus, potassium, and boron, were significantly affected by the sole and combined inoculation of AMF and PGPR (RB) strains. The PGPR (RB) and AMF proved highly significant for increasing rice growth by increasing the soil N contents. Higher contents of soil N were recorded in T11 (0.049%) and in T12 (0.0465%), which were 21.3% and 22.0% compared with the control, respectively. Minimum soil N contents were recorded at 0.0361% in T5 (PGPR2 = RB2) and T7 (PGPR3 = RB3), which were non-significant with each other. The comparison between the PGPR strains showed that PGPR5 (RB5) showed a remarkable increase in soil N contents, followed by PGPR1 and PGPR4, whereas PGPR2 (RB2) and PGPR3(RB3) inoculation reduced the soil N contents compared with the control treatment (Figure 2A).

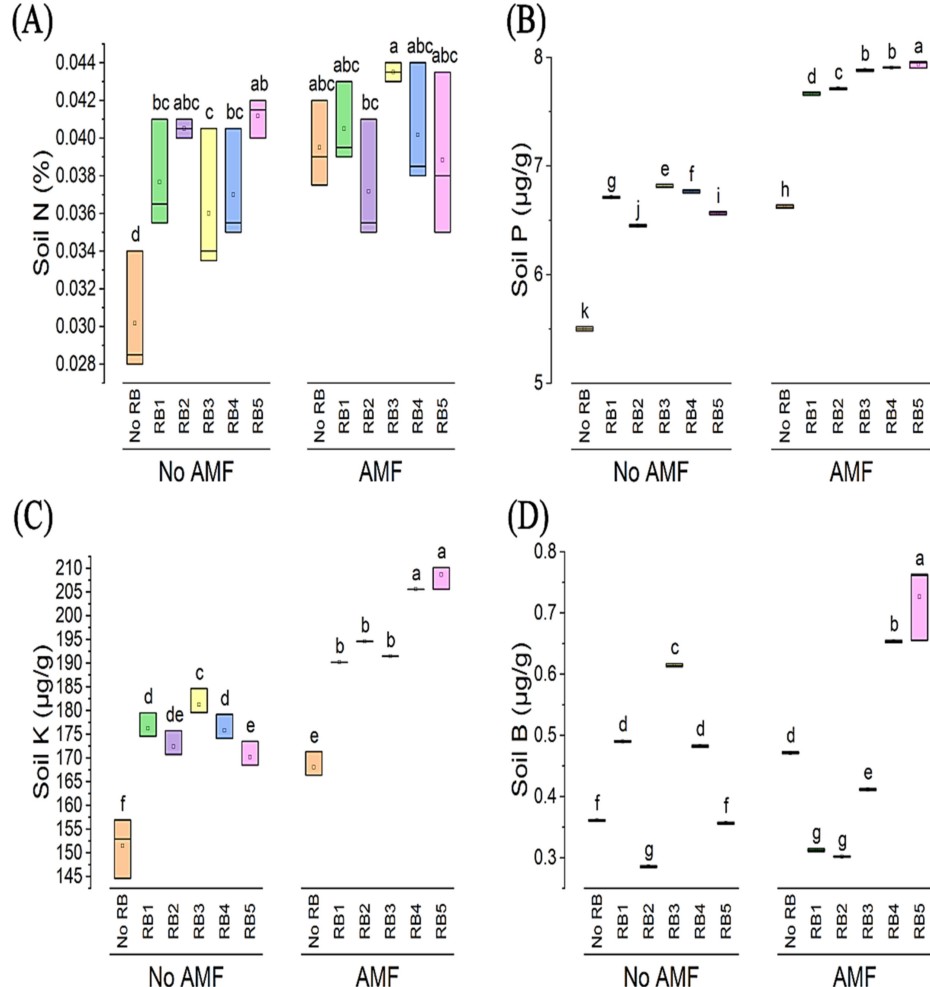

**Figure 2.** Effect of various strains of PGPR (RB1 = PGPR1, RB2 = PGPR2, RB3 = PGPR3, RB4 = PGPR4, RB5 = PGPR5) and AMF inoculation on soil nitrogen (**A**), phosphorus (**B**), potassium (**C**), and boron (**D**) contents. The bars having different letters indicate significance from each other at *p* < 0.05.



The AMF and PGPR strains' inoculation significantly affected the P availability in the soil. Statistical analysis showed that AMF and PGPR sole and combined inoculation significantly increased the P availability in the soil. Maximum available P in the soil was recorded at 6.39% in T12 (AMF + PGPR5), followed by 6.87% in T11 (PGPR5) and 6.7% in T10 (AMF + PGPR4), where T11 and T10 provided non-significant differences between each other. Minimum soil P availability was observed in the control treatment at 4.5%, as there was no fertilization of P practiced. AMF and PGPR5 remarkably increased the P availability in the soil, followed by PGPR4 and PGPR2. Sole application of PGPR1 reduced the P availability, whereas, when combined AMF and PGPR1 were applied, it increased the P availability. The PGPR2 provided contradicting results as their combined application reduced the P concentration in the soil, but sole PGPR2 inoculation increased the P availability up to 10% (Figure 2B).

Extractable K concentration in the soil was affected by the inoculation of AMF and various PGPR strains. It was observed that PGPR have the ability to increase the availability of K in the soil solution and enable its uptake by plant roots. Results showed that PGPR5 had a greater ability to increase the soil K content in the soil compared with other PGPR strains and the control treatment. Maximum soil K contents were shown in T12 (PGPR5 + AMF) (208.87%), followed by T11 (PGPR5). The PGPR5 had the ability to increase the mobility of K in the soil and that ability was increased by up to 10–15% when applied with AMF. It was reported that the minimum soil K was observed in T7 (PGPR3), followed by T6 (PGPR2 + AMF), which was 108% less compared with the highest K availability and 50% less compared with the control treatment (T1) (Figure 2C).

Boron concentration in the soil was profoundly ($p \leq 0.05$) influenced by the sole and combined application of PGPR strains and AMF. The results showed that the maximum soil boron was measured in T12 (AMF + PGPR5) at 0.72%, followed by 0.687% T11 (PGPR5), which was 47.22% more compared with the control treatment. The minimum soil boron was 0.26%, observed in the treatments receiving PGPR1, AMF + PGPR3, and PGPR4 applications. The PGPR3 and PGPR4 did not increase the soil boron extractability, even when co-inoculated with AMF. PGPR5, PGPR1, and PGPR2 provided a better result in increasing the soil B contents. In comparison with the PGPR and AMF results, it was analyzed that AMF had more ability to increase the soil B contents compared with the PGPR strains (Figure 2D).

### 3.3. Plant Leaf Analysis

The AMF and PGPR inoculation imparted a significant effect on increasing the N contents in the rice leaves, viz., plant leaf nitrogen, phosphorus, potassium, and boron. The accumulation of N in the plant tissues was the maximum (7.27%) in T12, which contains AMF and PGPR5, followed by T11 (6.85), which received a sole application of PGPR5. The PGPR-inoculated plants showed 10–40% more accumulation of N in the plant tissues. The PGPR3 and PGPR4 could not increase the N uptake by the plant roots. Similarly, AMF-inoculated treatments showed more accumulation of N in the rice tissue compared with non-inoculated plants. The minimum concentration of N in the plant tissues was 3.74%, observed in PGPR4 treatments either sole or when combined with AMF. It was found that the PGPR5 strain was much more effective in increasing the plant N concentration (Figure 3A). The AMF-inoculated plants showed more accumulation of P in the rice plants and the maximum P concentration in the plant tissues was 0.149%, analyzed in PGPR5-inoculated plants either sole or combined with AMF. The minimum P concentration was noted in the T1 control at 0.033% and T9 (PGPR4), which was 122% less than the highest P accumulation. PGPR5 proved much more effective compared with other strains' responses to P accumulation, followed by PGPR1 (Figure 3B). The maximum K uptake was observed in T12 (1.98%), i.e., AMF + PGPR5-inoculated rice plants, which were 54 % more than the control. The minimum K concentration in the plant tissue was 1.07% in PGPR2 + AMF-inoculated rice plants. Sole application of AMF also increased the K accumulation in the rice tissues, which was 38% less than the control treatment (Figure 3C).

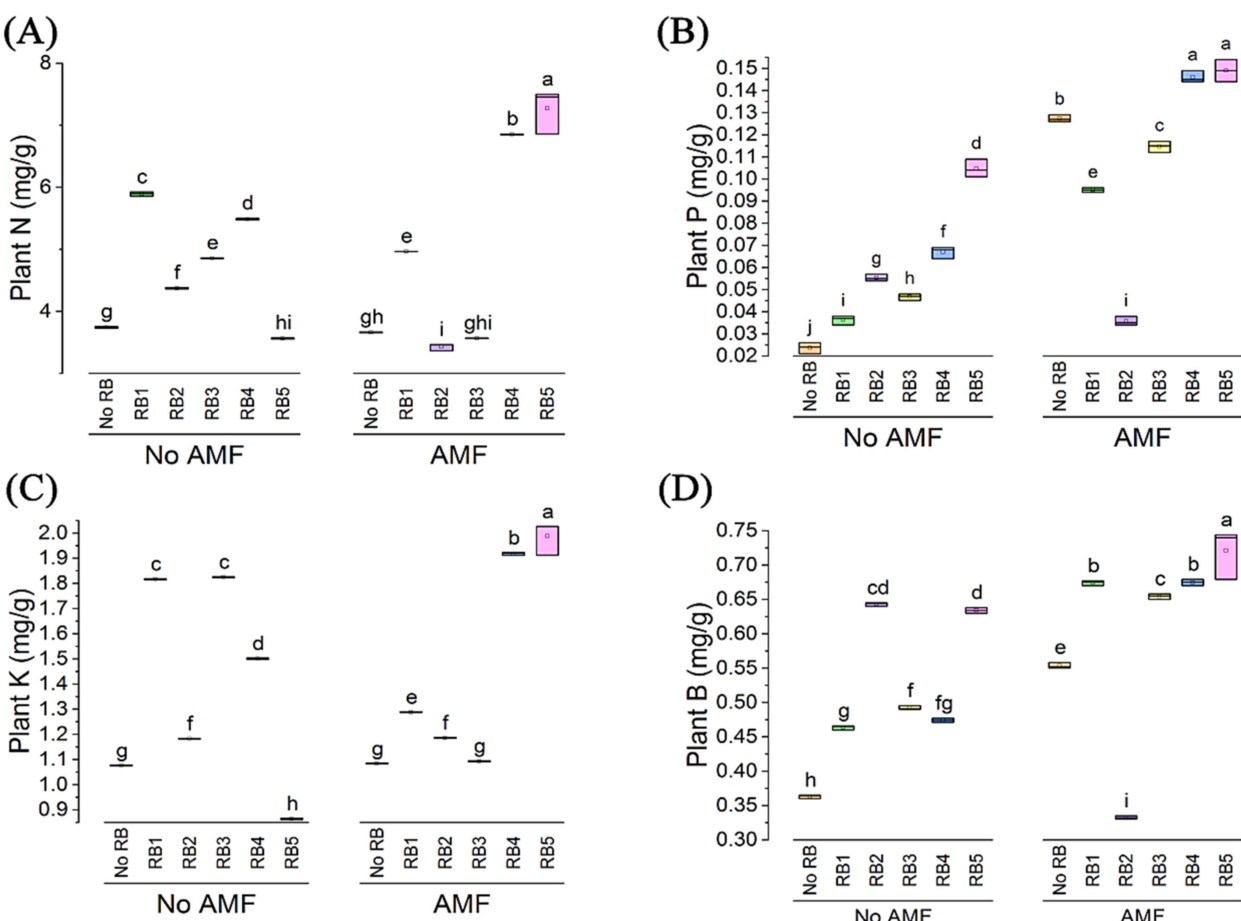

**Figure 3.** Effect of various strains of PGPR (RB1 = PGPR1, RB2 = PGPR2, RB3 = PGPR3, RB4 = PGPR4, RB5 = PGPR5) and AMF inoculation on nitrogen (**A**), phosphorus (**B**), potassium (**C**), and boron (**D**) contents in leaf tissues of rice. The bars having different letters indicate significance from each other at *p* < 0.05.

Boron concentration in the plant tissues was significantly (*p* ≤ 0.05) influenced by the application of AMF and PGPR strains. Maximum B uptake was observed in T12 (0.674%), i.e., AMF + PGPR5-inoculated rice plants, which were 38 % higher than the control. The minimum B concentration in the plant tissue was 0.363% in PGPR2 + AMF-inoculated rice plants (Figure 3D). It was observed that PGPR$_5$ respondedto plant nutrition much better than all other PGPR strains.

*3.4. Morpho-Biochemical Characteristics*

The AMF and PGPR inoculation imparted a substantial impact on raising the morpho-biochemical characters, viz., plant height, spike length, fresh weight, and dry weight, of the rice plant. The height of the rice plants was maximized up to 92 cm in T12, which contained AMF and PGPR$_5$, followed by T11 (89 cm), which received a sole application of PGPR5. The PGPR-inoculated plants showed 10–17% more plant height compared with the control plants. The control plants showed stunted growth. The response of the PGPR5 strain was much more effective in increasing plant growth compared with other treatments (Figure 4A).

The combined effect of AMF and PGPR strains on the length of the spikes of rice plants was significant. The spike length was maximized up to 26.1 cm in T12, which contained AMF and PGPR5, followed by T11 (25.5 cm), which received a sole application of PGPR$_5$. The PGPR-inoculated plants showed 30–40% tall spikes compared with the control

plants. The sole application of PGPR seemed to be more effective in increasing the length of spikes (Figure 4B).

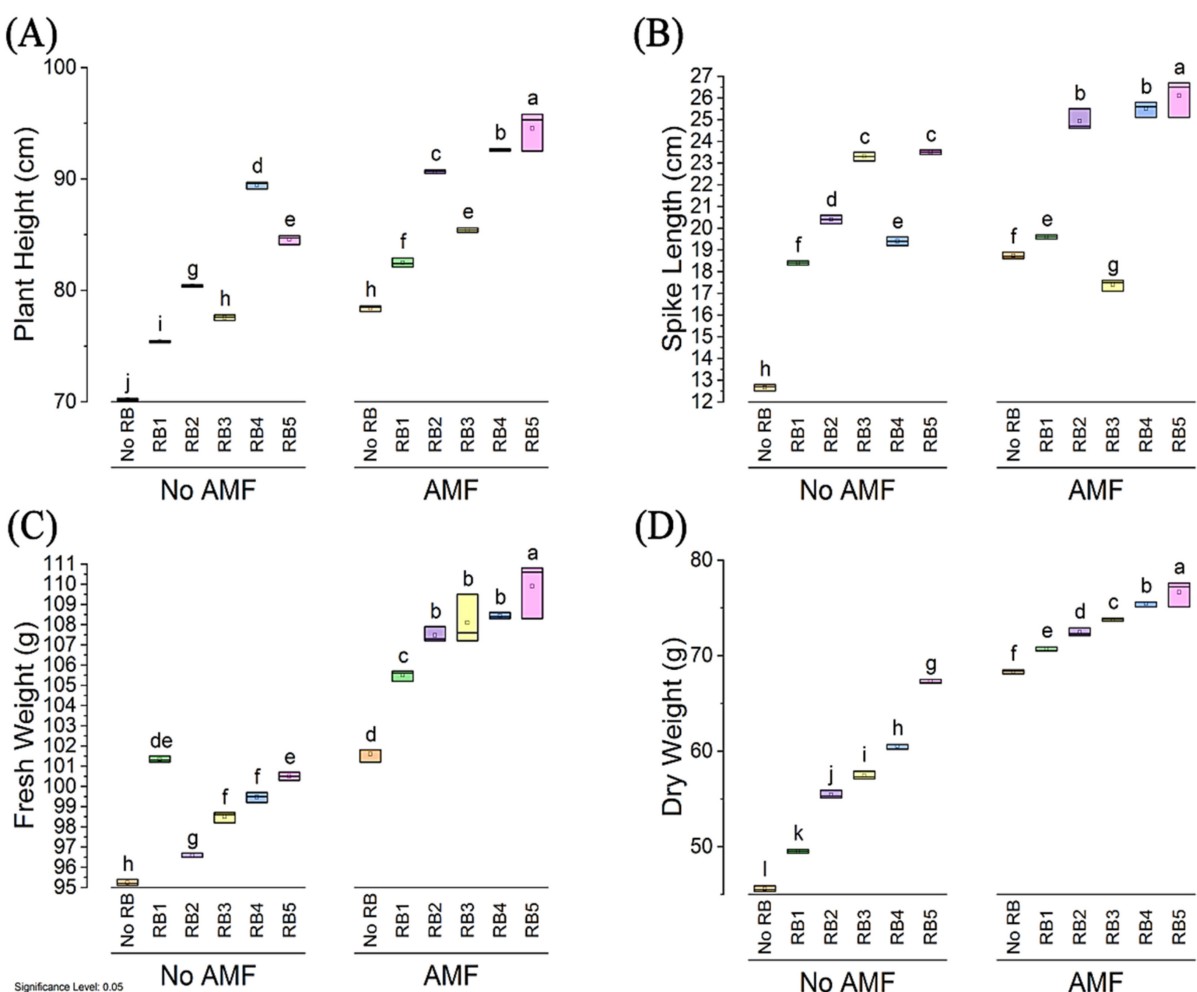

**Figure 4.** Effect of various strains of PGPR (RB1 = PGPR1, RB2 = PGPR2, RB3 = PGPR3, RB4 = PGPR4, RB5 = PGPR5) and AMF inoculation on the plant height (**A**), spike length (**B**), fresh weight (**C**), and dry weight (**D**) of rice. The bars having different letters indicate significance from each other at *p* < 0.05.

The maximum interactive effect of PGPR and AMF on dry weight was 76.63 in T11, followed by 73 g in T12. The control treatment produced a minimum dry weight of the plants. Rice growth and dry weight were positively influenced by the PGPR inoculation. Inoculated rice plants produced more dry mass than non-inoculated plants. The PGPR-inoculated rice plants also responded better than non-inoculated or control plants, but PGPR5 was most effective in increasing the dry weight of the rice crop (Figure 4D).

*3.5. Root Length, 1000-Grain Weight, No. of Tillers and No. of Spikes*

The maximum 1000-grain weight was observed with the PGPR5 sole inoculation of the rice plants. The highest 1000-grain weight was 21.66 g, which was 38% more than the control, which produced a minimum weight of 15.24 g (Figure 5A). Root length was remarkably increased with microbial inoculation. The PGPR responded more in increasing the root length than the AMF inoculation. The maximum root length was 25.8 cm in the T12 rice plants that received PGPR5 and AMF, 74% more than the control (Figure 5B). The tiller's production was significantly affected by the PGPR and AMF inoculation. The AMF and PGPR inoculation increased the production of tillers in rice plants. The maximum

number of tillers was recorded as 18 in T12 (AMF + PGPR5).The minimum number of tillers was recorded as 4.3 in the control. However, it was analyzed that the PGPR strain's effectiveness improved the tiller's production more than others (Figure 5C).

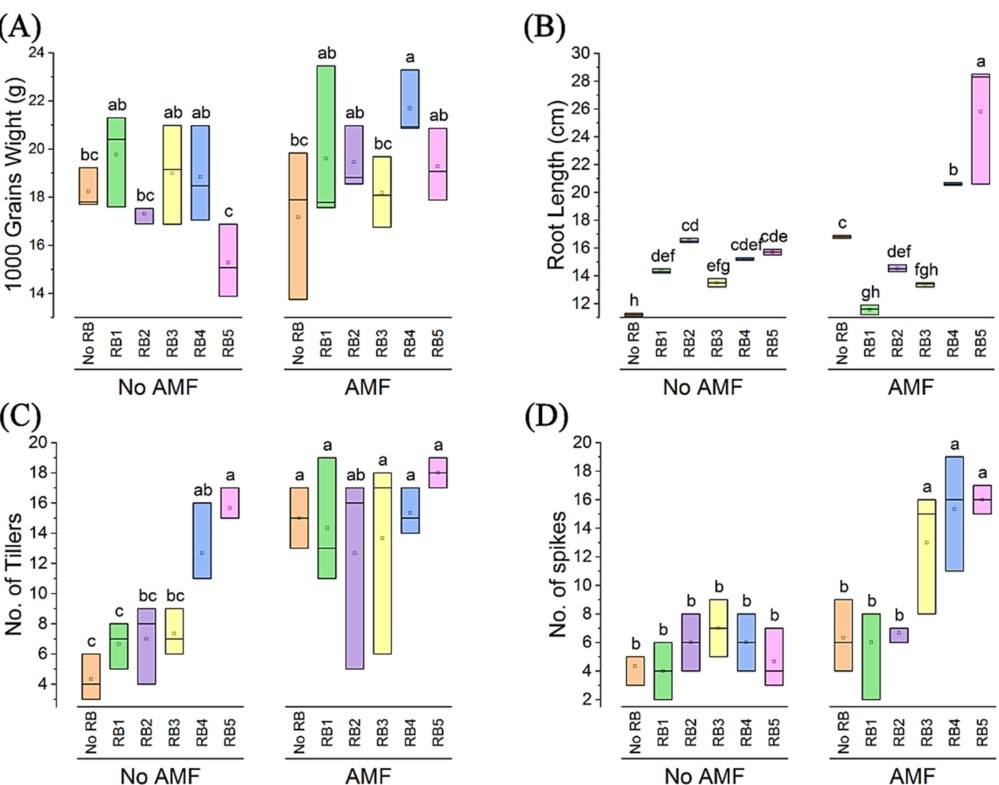

**Figure 5.** Effect of various strains of PGPR (RB1 = PGPR1, RB2 = PGPR2, RB3 = PGPR3, RB4 = PGPR4, RB5 = PGPR5) and AMF inoculation on the 1000-grain weight (**A**), plant root length (**B**), no. of tillers (**C**), and no. of spikes (**D**). The bars having different letters indicate significance from each other at $p < 0.05$.

### 3.6. Pearson's Correlation among the Variables and Principal Component Analysis

The RB showed a significant positive correlation with soil N, P, K, B, plant height, spike length, dry weight, no. of tillers, and no. of spikes. However, RB showed a significant negative correlation with soil pH, i.e., as the bacterial population increased, the soil pH decreased. Similarly, AMF was significantly positive in correlation with soil EC, OM, N, P, K, plant phosphorus, plant boron, plant height, dry weight, root length, no. of tillers, and no. of spikes (Figure 6). Both PC analyses depicted a 50.3% total variation. Variations were shown in the disease severity at different locations. The loading plots demonstrated the relationships between the disease severities at different locations, with a <90° angle of vectors positively correlated and a >90° angle of vectors not correlated (Figure 7).

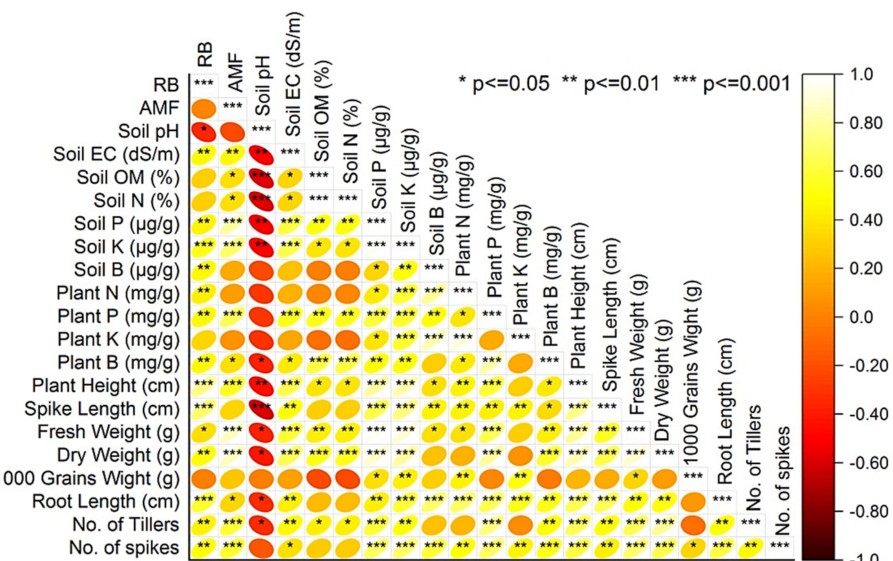

**Figure 6.** Pearson correlation among the variables and their interaction.

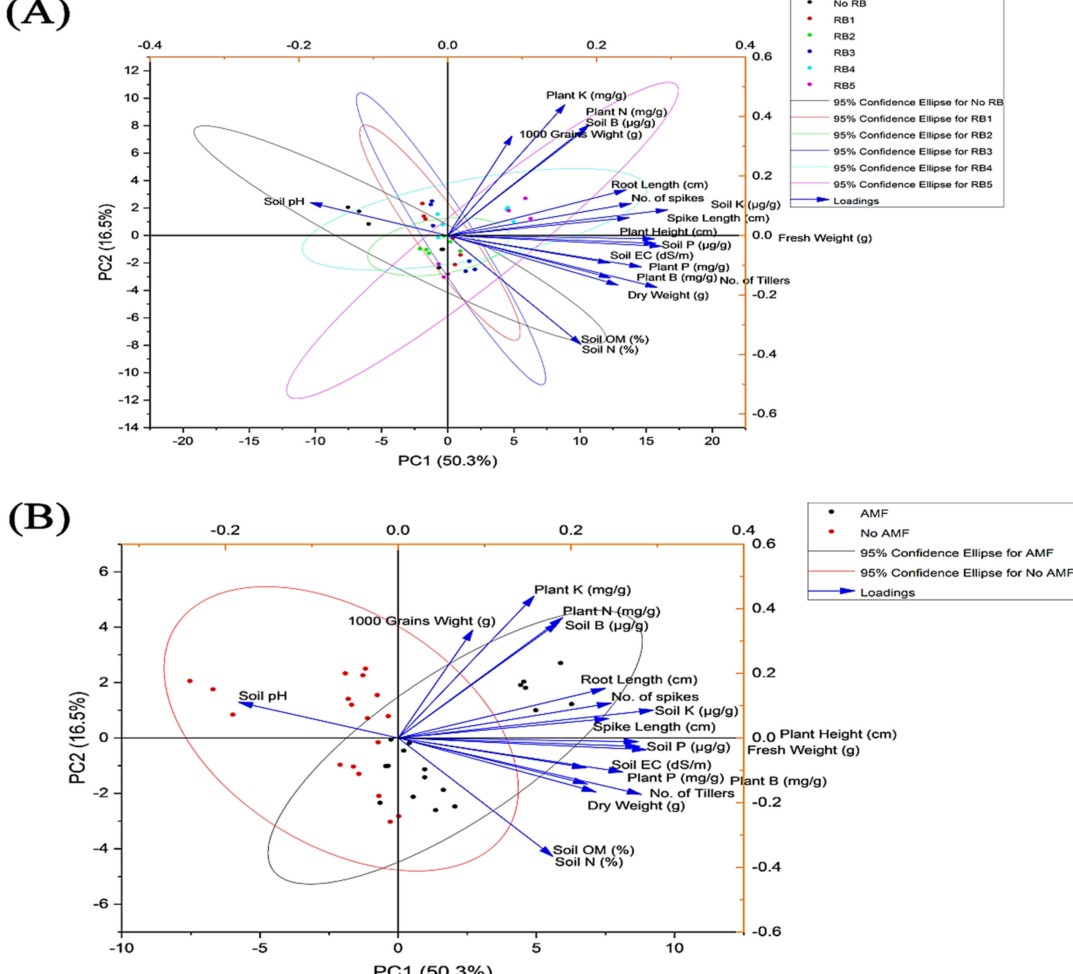

**Figure 7.** Principle component loading plots and scores of principal component analysis of different parameters of plants indicate that only one parameter has non-significant impact among all, whereas the sixteen parameters are showing significant observations that all the treatments had a positive impact on the increase of yield and mitigation of abiotic stress and increase in fecundity.

## 4. Discussion

The results indicated that the soil pH was decreased by up to one unit with the application of AMF. The decrease in soil pH was noted in treatment T4 (AMF + PGPR1) and T8 (AMF + PGPR3) strains. Another experiment showed that the soil pH was not remarkably affected by the AMF inoculation [45]. Similar to our results, the authors of thr reference [46,47] also found that the PGPR *Klebsiella pneumonia* (K5) strain decreased the soil pH to 5.4. Additionally, it was noted that the pH of the experimental orchard soil was dropped from 6.7 to 5.6–6.0 after the application of rhizobacteria. These findings may be explained by the fact that rhizobacterial colonization in soil lowers pH as a result of the synthesis of organic acids as a secondary metabolite, which, for example, can result in better soil conditions for raspberry cultivation [48]. Meanwhile, a single application of PGPR1 and PGPR4 enhanced the soil pH more than the control. Previous research [49] reported that bio-inoculation with the strain Cd-02 (*Cupriavidus* sp.) raised the soil pH in the culture medium from 7.40 to 6.68. In addition, the authors of the reference [49] reported that the soil was positively correlated with the PGPR strain.

The inoculation of sole and combined PGPR and AMF reduced the organic matter concentration in the soil. The ability of soil microbes to support material circulation, energy movement, nutrient transformation, organic matter decomposition, and other ecosystem-related biochemical processes was crucial for the functioning of the soil ecosystem [50,51]. The application of microbes, either PGPR or AMF, caused the rapid decomposition of organic matter by microbes. This was brought on by the plants' increased acidification, chelation, and exchange processes, which increased the mineralization of the organic materials [47]. Additionally, the use of AMF for plant growth in different biological habitats can significantly enhance organic culture for growth promotion and yield maximization [52]. AMFs are involved in soil organic matter decomposition through hydrolytic and oxidative activities or they assist in organic matter disintegration by stimulating the functioning of free-living saprotrophic organisms [53,54].

Overall results indicated that both PGPR and AMF can increase the soil N contents for promoting plant growth. AMF-inoculated plants displayed superior mineral nutritional status (N, P, K, and B) than non-inoculated plants, according to previous studies. The potential concentrations of N, P, K, and B in the soil and in the plants increased after PGPR and AMF inoculation [55,56]. Roots and hyphae in the soil were arranged spatially, suggesting that AMF might improve the re-absorption of nutrients lost by root exudation. The authors of [56] compared seeds that were not inoculated with AMF with the application of AMF and PGPR on the rice seeds, which showed increased yield and growth and also improved the chemical properties of the soil [57]. The use of AMF on the rice also showed a significant increase in growth under stress conditions and reduced metal toxicity in rice plants [58]. The present results showed that the inoculation of PGPR improved the available P contents in the soil solution. Similar results were reported by [59], who reported that in comparison to a single inoculation, the simultaneous inoculation of PGPR and AMF improved nutrient uptake and production of several crops in both normal and stressful settings. Furthermore, PGPR inoculation increased K availability and absorption. The decreased pH of the soil rhizosphere solution and the higher mineralization of the organic complex can be attributed to the increased availability of minerals in the soils as a result of bacterial applications [60]. Similar results also reported that PGPR strains, e.g., *Bacillus cereus* and *Pseudomonas* species, enhanced N, P, and K contents were 26%, 16%, and 31% in rice plants, respectively [60]. In the present study, AMF + PGPR maximized the B uptake in rice by 38% compared with the control T1. Previous research [61], observed that *Bacillus pumilus* inoculation inhibited plant growth and enhanced B uptake. Studies showed that AMF inoculation significantly increased plant biomass while reducing shoot B concentrations.

Research from the past has revealed that AMF and PGPR inoculation increased the levels of N, P, K, and B in soil and plants. In an experiment, the co-application of PGPR and rock phosphate significantly increased shoot dry matter; shoot N, P, K, Zn, and Mn

uptake [61]. The PGPR-inoculated plants showed 10–40% more accumulation of N in the plant tissues. Similarly, AMF-inoculated treatments showed more N accumulation in the rice tissue than in non-inoculated plants. Studies showed that PGPR improved plant growth, enhanced germination rate, and improved dry biomass in rice and wheat, which attributed to improved plant nutrition and photosynthesis rates [62,63]. In another study, PGPR (*Bacillus megaterium* M3, *Bacillus subtilis* OSU142, *Azospirillum brasilense* Sp245, and *Raoultella terrigena*) inoculation positively affected the dry weight and physiological parameters searched in both species in wheat and barley plants [64,65]. Plant growth parameters such as root, shoot length, number of tillers, and spike length were improved with microbial inoculation, as observed in the present study [66,67].

## 5. Conclusions

It is evident from the current research that soil conditions and growth attributes of the treated rice plants with the locally collected PGPR strains and AMF in the local experimental area significantly improved with the application of PGPR and AMF treatments. An application of a single treatment or its combined use promoted plant growth and development, which is a sign that the soil was in good health. Plant spike length and the height of the rice plants increased significantly after inoculation with AMF and PGPR. Hence, it is clear from the experiments conducted that the local application of AMF and PGPRs improved the nutrient contents in the soil and significantly enhanced their uptake in rice plants. This type of treatment can be very much effective for soil reclamation and plant yields for future studies and implementation for the stakeholders.

**Author Contributions:** Conceptualization, M.S., S.A.H.N. and M.J.R.; data curation, M.N.H.A.A.; formal analysis, S.A.H.N., M.J.R., M.R., W.A., Z.H. and M.Z.H.; funding acquisition, M.J.R. and S.A.H.N.; investigation, M.S., S.A.H.N., M.N.H.A.A., M.Z.H. and U.F.; methodology, M.S., S.A.H.N. and M.J.R.; project administration, D.C., M.J.R. and S.A.H.N.; resources, S.A.H.N.; software, M.J.R. and S.A.H.N.; supervision, M.M., M.A.-S., S.N. and M.F.H.; validation, M.J.R.; visualization, M.J.R. and M.S.; writing—original draft, M.S., M.N.H.A.A. and S.A.H.N., writing—review and editing, D.C., W.A., Z.H., M.Z.H., U.F., M.F.H., M.M., M.A.-S., S.N., M.J.R. and S.A.H.N. All authors have read and agreed to the published version of the manuscript.

**Funding:** Research and APC was funded by Chun Delai (22JR5RM207) through Natural Science Foundation of Gansu Province, China and Mahmoud Moustafa, Mohammed Al-Shehri and Sally Negm (KKU-IFP2-H-26) by the Ministry of Education in Kingdom of Saudi Arabia.

**Data Availability Statement:** Not applicable.

**Acknowledgments:** The authors are thankful for Natural Science Foundation of Gansu Province for funding this work through the Grant No. (22JR5RM207) and authors extend their appreciation to the Ministry of Education in KSA for funding the research work through the Project No. (KKU-IFP2-H-26).

**Conflicts of Interest:** The authors declare no conflict of interest. The funders had no role in the design of the study; in the collection, analyses, or interpretation of data; in the writing of the manuscript; or in the decision to publish the results.

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
