# Peer review of "Plant Growth Promoting Rhizobacteria (PGPR) and Arbuscular Mycorrhizal Fungi Combined Application Reveals Enhanced Soil Fertility and Rice Production"

_agronomy, doi:10.3390/agronomy13020550_

Round 1

Reviewer 1 Report

1) line 50 anedible -> an edible

2) line 50 andgrassy -> and grassy

3) line 72 importantgenera -> important genera

4) line 74 Arbuscularmycorrhizal -> Arbuscular mycorrhizal (and Why is it italicized?)

5) line 78 the pests’release -> the pests’ release

6) line 83 turnoverrate -> turnover rate

7) line 91 responses,when -> responses, when

8) line 119-120 thiophanatemethyl -> thiophanate-methyl

9) line pourswhere -> pours where

10) line 155 purple.N -> purple. N

11) line 164 added.Then -> added. Then

12) line 167 1,2,3,4 -> 1, 2, 3, 4

13) line 180 What’s man no. 41 -> Whatman No. 41

14) line 188 What’sman No. 40 paper. -> Whatman No. 40 filter paper

15) line 189 azome-thine-H -> azomethine-H

16) line 207 them.For -> them. For

17) line 218 boric acid.Then -> boric acid. Then

18) line 220 process.The -> process. The

19) line 228 1.Following -> 1. Following

20) line 236 h.The -> h. The

21) line 238 measurement.Flame -> measurement. Flame

22) line 240 For Boron;aaccording -> For Boron; according

23) line 243 furnace.After -> furnace. After

24) line 246 clods.The -> clods. The

25) line 247 What’s man No. filter -> Whatman No. OO filter paper (number needed)

26) line 254 parameters:Plant -> parameters: Plant

27) line 258 spikeswere -> spikes were

28) line 265 level.Pearson -> level. Pearson

29) line 270 unit.Maximum -> unit. Maximum

30) line 277 (Figure 1 A).The -> (Figure 1 A). The

31) line 281 Whereashigher -> Whereas higher

32) line 283 microbes.Minimum -> microbes. Minimum

33) line 296 Thecomparisn -> The comparison

34) line 298 Whereas, -> Whereas

35) line 324 Whereas, -> Whereas

36) line 344 (Figure 3 A).The -> (Figure 3 A). The

37) line 345 andmaximum -> and maximum

38) line 373 plantswas -> plants was

39) line 373-374 containedAMF -> contained AMF

40) line 379 Whereas, -> Whereas

41) line 389 weremaximum -> were maximum

42) line 410 PCanalyses -> PC analyses

43) line 429 Cupriavidua sp. -> Cupriavidus sp.

44) line 430 [50],reported -> [50], reported

45) line 436 [49].Whereas, -> [49]. Whereas

46) line 450 reference[56]in -> reference [56] in

47) line 451 AMF,application -> AMF, application

48) line 451 increasedyield -> increased yield

49) line 452 [57].Use -> [57]. Use

50) line 458 PGPRinoculation -> PGPR inoculation

51) line 461 Bacilluscereus -> Bacillus cereus

52) line 464 B. pumilus -> Bacillus pumilus

53) line 470 Mnuptake -> Mn uptake

54) line 475 Azospirillumbrasilense -> Azospirillum brasilense

55) line 476 Raoultellaterrigena -> Raoultella terrigena

56) line 486 Spikelength -> Spike length

57) In addition to the 56 I mentioned, there are too many space errors to count.

58) Be sure to check all of the references cited, as some scientific names are not in italics.

59) The contents of the first paragraph of the introduction and the contents that follow should be rich in the explanation. The last paragraph of the current authors' introduction is a description of PGPR and AMF, listing the applied results of PGPR and AMF in previous studies, and then one sentence related to the originality of the authors' research (Different strains of PGPR have different modes of action. ) only appeals. As of the current situation, it seems that the authors are saying that the authors only changed the material in the previously known content, so the originality is greatly reduced.

60) In the conclusion, it is necessary to describe what the authors' research results uncovered that were not previously known.

Author Response

Point by point response to Reviewer-I Comments and Suggestions

1) line 50 anedible -> an edible

Response: Thank you for your valuable comment, the correction has been made and highlighted in track changes.

2) line 50 andgrassy -> and grassy

Response: Thank you for your valuable comment, the correction has been made and highlighted in track changes.

3) line 72 importantgenera -> important genera

Response: Thank you for your valuable comment, the correction has been made and highlighted in track changes.

4) line 74 Arbuscularmycorrhizal -> Arbuscular mycorrhizal (and Why is it italicized?)

Response: Thank you for your valuable comment, the correction has been made and highlighted in track changes. Normally AMF is written in italics as this is fungi and described its scientific nature.

5) line 78 the pests’release -> the pests’ release

Response: Thank you for your valuable comment, the correction has been made and highlighted in track changes.

6) line 83 turnoverrate -> turnover rate

Response: Thank you for your valuable comment, the correction has been made and highlighted in track changes.

7) line 91 responses,when -> responses, when

Response: Thank you for your valuable comment, the correction has been made and highlighted in track changes.

8) line 119-120 thiophanatemethyl -> thiophanate-methyl

Response: Thank you for your valuable comment, the correction has been made and highlighted in track changes.

9) line pourswhere -> pours where

Response: Thank you for your valuable comment, the correction has been made and highlighted in track changes.

10) line 155 purple.N -> purple. N

Response: Thank you for your valuable comment, the correction has been made and highlighted in track changes.

11) line 164 added.Then -> added. Then

Response: Thank you for your valuable comment, the correction has been made and highlighted in track changes.

12) line 167 1,2,3,4 -> 1, 2, 3, 4

Response: Thank you for your valuable comment, the correction has been made and highlighted in track changes.

13) line 180 What’s man no. 41 -> Whatman No. 41

Response: Thank you for your valuable comment, the correction has been made and highlighted in track changes.

14) line 188 What’sman No. 40 paper. -> Whatman No. 40 filter paper

Response: Thank you for your valuable comment, the correction has been made and highlighted in track changes.

15) line 189 azome-thine-H -> azomethine-H

Response: Thank you for your valuable comment, the correction has been made and highlighted in track changes.

16) line 207 them.For -> them. For

Response: Thank you for your valuable comment, the correction has been made and highlighted in track changes.

17) line 218 boric acid.Then -> boric acid. Then

Response: Thank you for your valuable comment, the correction has been made and highlighted in track changes.

18) line 220 process.The -> process. The

Response: Thank you for your valuable comment, the correction has been made and highlighted in track changes.

19) line 228 1.Following -> 1. Following

Response: Thank you for your valuable comment, the correction has been made and highlighted in track changes.

20) line 236 h.The -> h. The

Response: Thank you for your valuable comment, the correction has been made and highlighted in track changes.

21) line 238 measurement.Flame -> measurement. Flame

Response: Thank you for your valuable comment, the correction has been made and highlighted in track changes.

22) line 240 For Boron;aaccording -> For Boron; according

Response: Thank you for your valuable comment, the correction has been made and highlighted in track changes.

23) line 243 furnace.After -> furnace. After

Response: Thank you for your valuable comment, the correction has been made and highlighted in track changes.

24) line 246 clods.The -> clods. The

Response: Thank you for your valuable comment, the correction has been made and highlighted in track changes.

25) line 247 What’s man No. filter -> Whatman No. OO filter paper (number needed)

Response: Thank you for your valuable comment, the correction has been made and highlighted in track changes.

26) line 254 parameters:Plant -> parameters: Plant

Response: Thank you for your valuable comment, the correction has been made and highlighted in track changes.

27) line 258 spikeswere -> spikes were

Response: Thank you for your valuable comment, the correction has been made and highlighted in track changes.

28) line 265 level.Pearson -> level. Pearson

Response: Thank you for your valuable comment, the correction has been made and highlighted in track changes.

29) line 270 unit.Maximum -> unit. Maximum

Response: Thank you for your valuable comment, the correction has been made and highlighted in track changes.

30) line 277 (Figure 1 A).The -> (Figure 1 A). The

Response: Thank you for your valuable comment, the correction has been made and highlighted in track changes.

31) line 281 Whereashigher -> Whereas higher

Response: Thank you for your valuable comment, the correction has been made and highlighted in track changes.

32) line 283 microbes.Minimum -> microbes. Minimum

Response: Thank you for your valuable comment, the correction has been made and highlighted in track changes.

33) line 296 Thecomparisn -> The comparison

Response: Thank you for your valuable comment, the correction has been made and highlighted in track changes.

34) line 298 Whereas, -> Whereas

Response: Thank you for your valuable comment, the correction has been made and highlighted in track changes.

35) line 324 Whereas, -> Whereas

Response: Thank you for your valuable comment, the correction has been made and highlighted in track changes.

36) line 344 (Figure 3 A).The -> (Figure 3 A). The

Response: Thank you for your valuable comment, the correction has been made and highlighted in track changes.

37) line 345 andmaximum -> and maximum

Response: Thank you for your valuable comment, the correction has been made and highlighted in track changes.

38) line 373 plantswas -> plants was

Response: Thank you for your valuable comment, the correction has been made and highlighted in track changes.

39) line 373-374 containedAMF -> contained AMF

Response: Thank you for your valuable comment, the correction has been made and highlighted in track changes.

40) line 379 Whereas, -> Whereas

Response: Thank you for your valuable comment, the correction has been made and highlighted in track changes.

41) line 389 weremaximum -> were maximum

Response: Thank you for your valuable comment, the correction has been made and highlighted in track changes.

42) line 410 PCanalyses -> PC analyses

Response: Thank you for your valuable comment, the correction has been made and highlighted in track changes.

43) line 429 Cupriavidua sp. -> Cupriavidus sp.

Response: Thank you for your valuable comment, the correction has been made and highlighted in track changes.

44) line 430 [50],reported -> [50], reported

Response: Thank you for your valuable comment, the correction has been made and highlighted in track changes.

45) line 436 [49].Whereas, -> [49]. Whereas

Response: Thank you for your valuable comment, the correction has been made and highlighted in track changes.

46) line 450 reference[56]in -> reference [56] in

Response: Thank you for your valuable comment, the correction has been made and highlighted in track changes.

47) line 451 AMF,application -> AMF, application

Response: Thank you for your valuable comment, the correction has been made and highlighted in track changes.

48) line 451 increasedyield -> increased yield

Response: Thank you for your valuable comment, the correction has been made and highlighted in track changes.

49) line 452 [57].Use -> [57]. Use

Response: Thank you for your valuable comment, the correction has been made and highlighted in track changes.

50) line 458 PGPRinoculation -> PGPR inoculation

Response: Thank you for your valuable comment, the correction has been made and highlighted in track changes.

51) line 461 Bacilluscereus -> Bacillus cereus

Response: Thank you for your valuable comment, the correction has been made and highlighted in track changes.

52) line 464 B. pumilus -> Bacillus pumilus

Response: Thank you for your valuable comment, the correction has been made and highlighted in track changes.

53) line 470 Mnuptake -> Mn uptake

Response: Thank you for your valuable comment, the correction has been made and highlighted in track changes.

54) line 475 Azospirillumbrasilense -> Azospirillum brasilense

Response: Thank you for your valuable comment, the correction has been made and highlighted in track changes.

55) line 476 Raoultellaterrigena -> Raoultella terrigena

Response: Thank you for your valuable comment, the correction has been made and highlighted in track changes.

56) line 486 Spikelength -> Spike length

Response: Thank you for your valuable comment, the correction has been made and highlighted in track changes.

57) In addition to the 56 I mentioned, there are too many space errors to count.

Response: Thank you for your valuable comment, the correction has been made and highlighted in track changes in the complete manuscript. Actually it happens when it is opened in another version of the window e.g., if the authors are using office 7 and the file is opened to a next version then it will show such type of mistakes. Anyhow all the such mistakes has been rectified.

58) Be sure to check all of the references cited, as some scientific names are not in italics.

Response: Thank you for your valuable comment, the correction has been made about the scientific names of the microorganisms and highlighted in track changes. The citations has also been cross checked by the reference list and found correct.

59) The contents of the first paragraph of the introduction and the contents that follow should be rich in the explanation. The last paragraph of the current authors' introduction is a description of PGPR and AMF, listing the applied results of PGPR and AMF in previous studies, and then one sentence related to the originality of the authors' research (Different strains of PGPR have different modes of action. ) only appeals. As of the current situation, it seems that the authors are saying that the authors only changed the material in the previously known content, so the originality is greatly reduced.

Response: Thank you for your valuable comment, the research was conducted by using the described PGPR and AMF which are very much important to enhance the fertility of the soil and also helps to boost the production of the concerned crops against what these have been used, similar is the case in this study these were utilized and found excellent in the reclamation of the soil and boosting the production. 

60) In the conclusion, it is necessary to describe what the authors' research results uncovered that were not previously known.

Response: Thank you for your valuable comment, normally it happens that farming community prefers to use the synthetic fertilizer and other synthetic materials which not only destroy the environment and also promote the toxicity level of the soil. This is the uniqueness of the research that it was conducted by using the described PGPR and AMF which are very much important to enhance the fertility of the soil and also helps to boost the production of the concerned crops against what these have been used, similar is the case in this study these were utilized and found excellent in the reclamation of the soil and boosting the production. It is the conclusion that the organic amendments in the field will be of greater quality for the management of the biotic and abiotic stresses of the crop plants.

Reviewer 2 Report

Although the topic is of interest there are plenty of works using PGPRs and AMF in rice. Authors should express clearly the novelty of their work compared to the state of art in the topic.

Abstract is larger than the limit indicated in authors guidelines.

According do authors guidelines, the abstract should place the question addressed in a broad context and highlight the purpose of the study; but the abstract lacks the broad context or the question addressed. What effect is intended to study and under which context? could be the question used for this matter.

Abstract lacks the brief description of the methodology used in the experiment

How soil nutrient availability was measured or increased is not clear in the abstract

Keywords. Although AMF is a common abbreviation, it should be expanded for the keywords section.

Line 50. there  "anedible"and "andgrassy" should be separated "an edible" and "and grassy"

Line 51, 65. lower case should be consistent among the text.

Line 72. separate importantgenera

Line 74. separate Arbuscularmycorrhizal fungus and keep lowercase consistent.

Line 74. AMF are not parasites

Line 82. lower case should be onsistent

Introduction should incude previous works on PGPR and AMF combined as there are plenty information regarding the combination.

Authors can include the effect of AMF or PGPR on rice

MATERIALS

authors should state the species composition of the consortium as it is available from the producer. The comercial inoculum states that it not only contains AMF, but contains Trichoderma and other bacterias (at least 6). Please state, which species of Trichoderma and the rest of bacteria are present and in what quantity . Please not the correct names as the nomencalture is outdated in the webpage of the producer.

Please state how many propagules per species is used in those 10 gr used.

Please list the species of PGPRs and in which quantity (CFU or spores) are going to be used.

line 194. Nomenclature is confusing as there are two abbreviations for each treatment. It was not mentioned in the PGPR inoculation section.

Line 261. if there where only 3 replicates used, please state the normality tests.

Figure 1. is using yet another abreviation to refer to the treatmetns, previously was PGPR1,later T1 and now RB1 Please unify the treatment abbreviation.

The text has the same issues regarding treatment nomenclature and readers will be confused. Please unify the nomenclature

Discussion should focus on the results and explain the possible explanation that yielded them. The results can contribute to explain the changes observed

Please state clearly the experimental design and how they were analyzed and not only list the treatments.

Author Response

Point by point response to Reviewer-II Comments and Suggestions

Although the topic is of interest there are plenty of works using PGPRs and AMF in rice. Authors should express clearly the novelty of their work compared to the state of art in the topic.

Response: Thank you for your valuable comments, surely the current study provides uniqueness in itself as The symbiotic relationship of Plants to mycorrhiza is proven very effective and optimized approach to preserve nutrients for plants because using chemical fertilizers is hazardous to environment. Recently investigations were conducted to analyze the impact of PGPR inoculation and mycorrhizal fungi on soil chemical properties and rice plants. The results suggest the major correlation between PGPR inoculation and AMF activity for plant growth and development. The spike length and height of rice plants have been increased after the inoculation of AMF and PGPR. Hence, it was concluded that AMF and PGPR in the sole and/or combined application improved nutrient contents in soil and significantly enhanced their uptake in rice plants. 

Abstract is larger than the limit indicated in authors guidelines. According do authors guidelines, the abstract should place the question addressed in a broad context and highlight the purpose of the study; but the abstract lacks the broad context or the question addressed. What effect is intended to study and under which context? could be the question used for this matter.

Response: Thank you for your valuable comment, the abstract has been addressed and the omits has been made wherever it were necessary to omit word or sentence. The current study research question was either the plant growth-promoting Rhizobacteria (PGPR) and Arbuscular mycorrhizal fungus (AMF) perform their effect on soil fertility and rice growth? And as rice plants were inoculated to evaluate how AMF fungi and PGPR affect various aspects of soil and plants implicate in abiotic stress tolerance.

Abstract lacks the brief description of the methodology used in the experiment

Response: Thank you for your valuable comment, the correction has been made and the description of the experimental design and the methodology has been added in the abstract. The experiment was carried out in the controlled conditions in earthen pots in a completely randomized design with three replicates.

How soil nutrient availability was measured or increased is not clear in the abstract

Response: Thank you for your valuable comment, the various level of different micro and macro nutrients were measured by the soil and from the leaves of the crop in question and in this way the accumulation of the nutrients were estimated.

Keywords. Although AMF is a common abbreviation, it should be expanded for the keywords section.

Response: Thanks for your valuable suggestion, AMF has been added up in the key words list.

Line 50. there  "anedible"and "andgrassy" should be separated "an edible" and "and grassy"

Response: Thank you for your valuable comment, the correction has been made and highlighted in track changes in the main file.

Line 51, 65. lower case should be consistent among the text.

Response: Thank you for your valuable comment, the correction has been made and highlighted in track changes in the main file.

Line 72. separate importantgenera

Response: Thank you for your valuable comment, the correction has been made and highlighted in track changes in the main file.

Line 74. separate Arbuscularmycorrhizal fungus and keep lowercase consistent.

Response: Thank you for your valuable comment, the correction has been made and highlighted in track changes in the main file.

Line 74. AMF are not parasites

Response: Thank you for your valuable comment, the correction has been made and highlighted in track changes in the main file.

Line 82. lower case should be onsistent

Response: Thank you for your valuable comment, the correction has been made and highlighted in track changes in the main file.

Introduction should incude previous works on PGPR and AMF combined as there are plenty information regarding the combination.

Response: Thank you for your valuable comment, the previous stuides have been incorporated in the text to strengthen the introduction and discussion section and correction has been made and highlighted in track changes in the main file.

Authors can include the effect of AMF or PGPR on rice

Response: Thank you for your valuable comment, The results suggest the major correlation between PGPR inoculation and AMF activity for plant growth and development. The spike length and height of rice plants have been increased after the inoculation of AMF and PGPR. Hence, it was concluded that AMF and PGPR in the sole and/or combined application improved nutrient contents in soil and significantly enhanced their uptake in rice plants.

MATERIALS

authors should state the species composition of the consortium as it is available from the producer. The comercial inoculum states that it not only contains AMF, but contains Trichoderma and other bacterias (at least 6). Please state, which species of Trichoderma and the rest of bacteria are present and in what quantity . Please not the correct names as the nomencalture is outdated in the webpage of the producer.

Response: Thank you for your valuable comments, in our study for AMF inoculation, about 5 cm deep holes were made in the experimental pots, then about 10g mycorrhizal inoculum predominantly containing Glomus species along with 9 propagules (include the spores, hyphal fragments and root portions) of Gigaspora albida (Clonex® Root Maximizer; Bustan, Toronto, Canada) as inoculum was used in this study as was predominantly rich in Glomus species. The correction has been made in the main file in track changes.

Please state how many propagules per species is used in those 10 gr used.

Response: Thank you for your valuable comment, 9 propagules  (include the spores, hyphal fragments and root portions) of Gigaspora albida were used.

Please list the species of PGPRs and in which quantity (CFU or spores) are going to be used.

Response: Thank you for your valuable comment, Paenibacillus (PGPR 1), Rhizobium (PGPR 2), Bacillus (PGPR 3), Azotobacter (PGPR 4), and Pseudomonas (PGPR 5) were used as PGPR. Briefly, the solution of the sugar and water (0.25:1 ratio) was prepared. After that, in the solution of sugar and water 80mg of PGPR inoculum containing 106 cfu/ml calibrated at spectrophotometer of each was added in the current research.

line 194. Nomenclature is confusing as there are two abbreviations for each treatment. It was not mentioned in the PGPR inoculation section.

Response: Thank you for your valuable comment, the correction has been made and highlighted in track changes in the main file.

Line 261. if there where only 3 replicates used, please state the normality tests.

Response: Thank you for your valuable comment, the experiment was conducted in a completely randomized design with three replicates and the collected datasets were first normalized by using the origin software and excel software and then it was processed for analysis of variance and variability in the treatments were determined.

Figure 1. is using yet another abreviation to refer to the treatmetns, previously was PGPR1,later T1 and now RB1 Please unify the treatment abbreviation.

Response: Thank you for your valuable comment, the correction has been made in the figure legend of the figure 1 while the treatments T1 and etc are actually explained alongside their description.

The text has the same issues regarding treatment nomenclature and readers will be confused. Please unify the nomenclature

Response: Thank you for your valuable comment, the clarification of the PGPR or RB and the treatments has been rectified in the complete manuscript and now it will be easy for the readers to understand the treatments.  

Discussion should focus on the results and explain the possible explanation that yielded them. The results can contribute to explain the changes observed

Response: Thank you for your valuable comment, the correction has been made and highlighted in track changes in the main file.

Please state clearly the experimental design and how they were analyzed and not only list the treatments.

Response: Thank you for your valuable comment, the correction has been made in the manuscript and highlighted in track changes as the experiment was conducted in a completely randomized design with three replicates. This has been added up in the experimental design section.  

Reviewer 3 Report

Brief summary:

The manuscript Agronomy-2184349- entitled “Sustainable Management of Soil Fecundity to Improve Rice Production using Rhizobacteria and Arbuscular mycorrhizae” investigated the mono and the combined inoculation with Plant Growth-Promoting Bacteria (PGPB) and Arbuscular Mycorrhizae (AMF) on rice growth and soil fertility.  The work is interesting and provides further knowledge in the area. Valid procedures were used to conduct the experiment. The data handling is suitable. However, the manuscript presents some flaws in the quality of the presentation. See specific comments below. The English language should be revised for correctness and fluency. 

Specific comments:

-In the abstract section, it would be better if you provided more details about the working methodologies used in the study, at the same time is better to reduce some aspects of the results and keep just the main ones.

-Line 39 what do you mean by PGPR5? The same is in lines; 40,43, and 45. You should clarify it for readers as the abstract is an abbreviated image of what is written in the manuscript.

-The keyword should be representative elements of the work.  However, abiotic stress and salinity are untouched points in your work.

-The English language must be revised in whole the manuscript.

Introduction: The Introduction correctly places the study in the context. However, a description of the working hypothesis should be provided.

-Correct space in lines 67, 71, 76, and 80… verify throughout the manuscript.

-Line 86-87: “Growth rate of roots is effected due to Mycorrhizal association or does not affect xylem pressure” reformulate the phrase for more clarity and correct the word “effected”.

-Line 111: Reference 65 in the introduction part is misnumbered.

Materials and Methods: The authors described the used methods clearly. I would add missing points of some experiments:

-The location of the rice nursery should be provided.

-Glomus species were the main species in AMF consortium, however, no information about the representative species in PGPR consortium was provided.

-Line 126-128: “The roots of the plants were dipped into the solution from where bacterial strains were connected to the roots and then transplantation to into the pots was done”. have you performed some procedures to confirm the colonization of the root by PGPR?

-Line 130: “2.2. Soil collection and characterization of Soil pH, EC and organic matter”. It would be better if you simplify the title for soil analysis, you don’t have to mention all the analyzed parameters.

-I suggest you to start with soil analysis then the preparation of the inocula as you checked firstly the chemical and nutritional composition and health of the soil before pots filling.

-Move table 1 to the results part.

-You mentioned 5 types of PGPR, are there different types of biofertilizers used separately? What is the main strains in each type? And based on what you choose 5 PGPR strains and only one AMF?

-Line 116: “The rice nursery was grown and seven kg of soil was filled in earthen pots (10”×45”)”, and in line 195-196 “Rice seedlings were grown in sandy loam soil. Each pot (12 kg size) was filled with a mixture of 8 kg soil and 2 kg of sand”. The information is mixed about the quantity of soil in each pot!!

-Lines 254-255: “Agronomic parameters: Plant height, spike length, fresh weight, dry weight, 1000 grains weight, Root length, No. of tillers and No. of spikes”. simplify the title with morpho-biochemical characteristics.

Results and discussion: 

-In figures 1 to 5 in the caption; “Effect of various strains of PGPR and AMF inoculation on…”. The conditions indicated in the different figures are AMF and no AMF, there is no indication of PGPR strains used. Moreover, many results are presented with the same graphic type (5), try to diversify the graphical presentation of the results. In the first section, it would be better to present the soil parameters with a table.

-The authors correctly discussed the results from the perspective of previous studies. However, this section must be ameliorated especially in the first section, highlighting the role of pH decrease in the soil and the availability of nutrients for plants.

Conclusion: This section must be improved and the English language must be revised for correctness and fluency. Moreover, this section must be improved and future research directions should be added.

Author Response

Point by point response to reviewer-III Comments and Suggestions

Brief summary:

The manuscript Agronomy-2184349- entitled “Sustainable Management of Soil Fecundity to Improve Rice Production using Rhizobacteria and Arbuscular mycorrhizae” investigated the mono and the combined inoculation with Plant Growth-Promoting Bacteria (PGPB) and Arbuscular Mycorrhizae (AMF) on rice growth and soil fertility.  The work is interesting and provides further knowledge in the area. Valid procedures were used to conduct the experiment. The data handling is suitable. However, the manuscript presents some flaws in the quality of the presentation. See specific comments below. The English language should be revised for correctness and fluency. 

Response: Thank you for encouraging remarks and valuable comments, the corrections has been made in the manuscript and the English language editing has been done by the native English speaker to improve the quality of the article.

Specific comments:

-In the abstract section, it would be better if you provided more details about the working methodologies used in the study, at the same time is better to reduce some aspects of the results and keep just the main ones.

Response: Thank you for your valuable comments, the corrections has been made in the abstract section and highlighted in the track changes.

-Line 39 what do you mean by PGPR5? The same is in lines; 40,43, and 45. You should clarify it for readers as the abstract is an abbreviated image of what is written in the manuscript.

Response: Thank you for your comment, the concerned query has been addressed in the materials and methods section and deliberated in detail to understand it better, the names of the PGPRs are hereby motioned and all the queries related to the PGPR, AMF, Treatments, RB etc has been addressed at the proper places and highlighted in red.  

-The keyword should be representative elements of the work.  However, abiotic stress and salinity are untouched points in your work.

Response: Thank you for your valuable comments, the untouched words from the key words list has been omitted and the relevant terms are hereby added up in the list and highlighted in track changes.

-The English language must be revised in whole the manuscript.

Response: Thank you for your valuable comment, to improve the quality of the English services of a native English speaker has been borrowed and all the language mistakes has been addressed.

Introduction: The Introduction correctly places the study in the context. However, a description of the working hypothesis should be provided.

Response: Thank you for your valuable comments, the hypothesis has been added up at the end of the introduction section of the article.

-Correct space in lines 67, 71, 76, and 80… verify throughout the manuscript.

Response: Thank you for your valuable comments, the corrections has been made for the spaces in the complete manuscript and highlighted in track changes. 

-Line 86-87: “Growth rate of roots is effected due to Mycorrhizal association or does not affect xylem pressure” reformulate the phrase for more clarity and correct the word “effected”.

Response: Thank you for your valuable comments, the corrections has been made for the spaces in the complete manuscript and highlighted in track changes as Mycorrhizal association is a symbiotic relationship between plants and fungi where the fungi provide nutrients and water to the plant in exchange for sugars produced through photosynthesis. This relationship affects the growth rate of plant roots as it improves their ability to absorb nutrients and water, leading to increased root growth. On the other hand, xylem pressure, which refers to the pressure of water in the plant's xylem tissue, does not have a direct impact on the growth rate of roots.

-Line 111: Reference 65 in the introduction part is misnumbered.

Response: Thank you for your valuable comment, the correction in the reference has been made.

Materials and Methods: The authors described the used methods clearly. I would add missing points of some experiments:

-The location of the rice nursery should be provided.

Response: Thank you for your valuable comment, the location of the nursery preparation has been added up at the proper place.

-Glomus species were the main species in AMF consortium, however, no information about the representative species in PGPR consortium was provided.

Response: Thank you for your valuable comments, all the required information of both treatments has been given in the updated version of the article at proper places.

-Line 126-128: “The roots of the plants were dipped into the solution from where bacterial strains were connected to the roots and then transplantation to into the pots was done”. have you performed some procedures to confirm the colonization of the root by PGPR?

Response: Thank you for your valuable comment, the response of the plant as compared to the control highlights the effect of the PGPR and demonstrated clearly the impact of PGPR on rice plants.

-Line 130: “2.2. Soil collection and characterization of Soil pH, EC and organic matter”. It would be better if you simplify the title for soil analysis, you don’t have to mention all the analyzed parameters.

Response: Thank you for your valuable comment, the correction has been made in the article at the appropriate place

-I suggest you to start with soil analysis then the preparation of the inocula as you checked firstly the chemical and nutritional composition and health of the soil before pots filling.

-Move table 1 to the results part.

Response: Thank you for your valuable comment, the correction has been made in the article at the appropriate place.

-You mentioned 5 types of PGPR, are there different types of biofertilizers used separately? What is the main strains in each type? And based on what you choose 5 PGPR strains and only one AMF?

Response: Thank you for your valuable comment, yes five different types of PGPR (RB) were used in this study whose names has also been given in the material and methods section, these were used in the lephoized form not in the fertilizer form. Only one AMF was obtained from the concerned quarter whose information is now given in the text and was used in the experiment. 

-Line 116: “The rice nursery was grown and seven kg of soil was filled in earthen pots (10”×45”)”, and in line 195-196 “Rice seedlings were grown in sandy loam soil. Each pot (12 kg size) was filled with a mixture of 8 kg soil and 2 kg of sand”. The information is mixed about the quantity of soil in each pot!!

Response: Thank you for your valuable comment, the suggested changes has been rectified and amended at the proper places. 

-Lines 254-255: “Agronomic parameters: Plant height, spike length, fresh weight, dry weight, 1000 grains weight, Root length, No. of tillers and No. of spikes”. simplify the title with morpho-biochemical characteristics.

 Response: Thank you for your valuable comment, the correction has been made in the article at the appropriate place.  

Results and discussion: 

-In figures 1 to 5 in the caption; “Effect of various strains of PGPR and AMF inoculation on…”. The conditions indicated in the different figures are AMF and no AMF, there is no indication of PGPR strains used. Moreover, many results are presented with the same graphic type (5), try to diversify the graphical presentation of the results. In the first section, it would be better to present the soil parameters with a table.

Response: Thank you for your valuable comment, the query in comment has been addressed in the complete legends of the figures from 1 to 4 and the information’s has also been provided much clearer in the text regarding the treatments,

-The authors correctly discussed the results from the perspective of previous studies. However, this section must be ameliorated especially in the first section, highlighting the role of pH decrease in the soil and the availability of nutrients for plants.

Response: Thank you for your valuable comment, the results and the discussion section has been ameliorated from the perspective of the previous studies.

Conclusion: This section must be improved and the English language must be revised for correctness and fluency. Moreover, this section must be improved and future research directions should be added.

Response: Thank you for your valuable comment, the English editing service has been obtained by a native English speaker to improve the quality of the article.

Reviewer 4 Report

Artigo bem escrito. Aprovado

Author Response

Point by point response to Reviewer IV comments

Line 47: add mycorhical fungi in key words

Response: Thank you for your valuable comment, the suggested key word has been incorporated in the key words list.

Line 487-489: It would also be important to inform which nutrients obtained the best response

Response: Thank you for your valuable comment, surely plant will be more vigorous by maintaining a strong connection with AMF and roots will be better in position to obtain N, P, K, Ca, Mg etc to improve its health status and finally for a better yield.

Round 2

Reviewer 1 Report

What I said about 59) and 60) was not to persuade me. It was a request to explain the originality that only the authors know in an easy-to-understand way for readers to accept. It is a fact that you do not need to be a scientist to know that there are numerous mutant strains and that changing them will change the results. A sentence that explains the meaning of your research in an easy-to-understand manner is required.

for example:

Because there is a difference between the strains available in the environment the authors are in and the strains studied worldwide, experiments were conducted with strains that are easy to obtain locally to study how they affect local soil and plants.

By finding that the soil conditions of the experimental area are improved from A to B with the strains available in the authors' experimental area, your research will be useful within the same country with similar soil environments.

Author Response

Point by point response to Reviewer-I (Round-II) Comments and Suggestions

What I said about 59) and 60) was not to persuade me. It was a request to explain the originality that only the authors know in an easy-to-understand way for readers to accept. It is a fact that you do not need to be a scientist to know that there are numerous mutant strains and that changing them will change the results. A sentence that explains the meaning of your research in an easy-to-understand manner is required.

for example:

Because there is a difference between the strains available in the environment the authors are in and the strains studied worldwide, experiments were conducted with strains that are easy to obtain locally to study how they affect local soil and plants.

By finding that the soil conditions of the experimental area are improved from A to B with the strains available in the authors' experimental area, your research will be useful within the same country with similar soil environments.

For 59

Response: Thank you for your valuable comment, as per guidance, at 59. The following statement has been added up in the last paragraph of the introduction: Nature is the best source for any remedy or a problem faced by the agriculture, in order to deal with the soil problems and to improve the crop productivity, PGPR and AMF which are found in the soil as natural microbiome which were collected from the local soil vicinities in the fields and from the plants rhizosphere offer the best source of solution for the domestic soils for both soil problems and yield improvement. In PGPR there is a great diversity due to the presence of sexual reproduction among their populations through conjugation, transformation and transduction and they can play very positive role for soil health. Hence, experiments were carried out with the locally obtained strains of the PGPR and AMF to observe their efficacy for the domestic issue of the soil and productivity of the rice crop.

For 60

Response: Thank you for your valuable comment, as per guidance, at 60. The following statement has been added up in the conclusion section: It is evident from the current research that soil conditions and growth attributes of the treated rice plants with the locally collected PGPR strains and AMF in the local experimental area significantly improved by the application of PGPR and AMF treatments. Application of single treatment or its combined use promoted plant growth and development which is a sign that soil is in good health. Plant spike length and height of rice plants increased significantly after the inoculation of AMF and PGPR. Hence, it is clear from the experiments conducted that local application of AMF and PGPRs improved nutrient contents in soil and significantly enhanced their uptake in rice plants. This type of treatment can be very much effective for soil reclamation and plant yields for future studies and implementation for the stakeholders.

Reviewer 3 Report

Brief summary:

The manuscript Agronomy-2184349- entitled “Plant Growth Promoting Regulators (PGPR) and Arbuscular Mycorrhizal Fungi Combined Application Reveals Enhanced Soil Fertility and Rice Production” examined the effects of single and combined inoculation with Arbuscular Mycorrhizal Fungi (AMF) and Plant Growth-Promoting Bacteria (PGPB) on rice growth and soil fertility. The authors ameliorated the manuscript quality according to the reviewers’ suggestions. The English language was revised. I would like to ameliorate only minor aspects, see please specific comments.  

Specific comments

Line 143: Correct the reference numbering.

-I suggest revising the titles to clarify them for the readers; “2.2. Soil collection and characterization of Soil pH, EC, and organic matter”, “2.5. Plants Analysis: leaf Nitrogen, Phosphorus, Potassium, and Boron”, “2.6. Agronomic parameters: Plant height, spike length, fresh weight, dry weight, 1000 grains weight, Root length, No. of tillers and No. of spikes”.

Line 495: Replace “principle component analysis » with “principal component analysis”.

-Move table 1 to the results part.

-I suggest reformulating the conclusion part proving future research directions as perspectives.

Author Response

Point by Point Response to Reviewer-III (Round-II) Comments and Suggestions

Brief summary:

The manuscript Agronomy-2184349- entitled “Plant Growth Promoting Regulators (PGPR) and Arbuscular Mycorrhizal Fungi Combined Application Reveals Enhanced Soil Fertility and Rice Production” examined the effects of single and combined inoculation with Arbuscular Mycorrhizal Fungi (AMF) and Plant Growth-Promoting Bacteria (PGPB) on rice growth and soil fertility.

The authors ameliorated the manuscript quality according to the reviewers’ suggestions. The English language was revised. I would like to ameliorate only minor aspects, see please specific comments. 

Response: Thank you for your encouraging remarks, specific comments has also been addressed to improve the quality of the article. 

Specific comments

Line 143: Correct the reference numbering.

Response: Thank you for your valuable comment, the correction has been made and the reference [66] has been deleted from this place. As there is no need of this reference here. Further numbering has been checked carefully and rectified where it was appropriate.

-I suggest revising the titles to clarify them for the readers; “2.2. Soil collection and characterization of Soil pH, EC, and organic matter”, “2.5. Plants Analysis: leaf Nitrogen, Phosphorus, Potassium, and Boron”, “2.6. Agronomic parameters: Plant height, spike length, fresh weight, dry weight, 1000 grains weight, Root length, No. of tillers and No. of spikes”.

Response: Thank you for your valuable comment, the titles of the sub sections 2.2; 2.5; and 2.6 have been revised to make them easy for the readers to study and understand. They are highlighted in track changes.

Line 495: Replace “principle component analysis » with “principal component analysis”.

Response: Thank you for your valuable comment, the correction as mentioned above has been made at two different positions and highlighted in track changes.

-Move table 1 to the results part.

Response: Thank you for your valuable comment, the correction has been made and the table move to results section and showed in track changes.

-I suggest reformulating the conclusion part proving future research directions as perspectives.

Response: Thank you for your valuable comment, the conclusion part has been reformulated as per the suggestion, now it is “It is evident from the current research that soil conditions and growth attributes of the treated rice plants with the locally collected PGPR strains and AMF in the local experimental area significantly improved by the application of PGPR and AMF treatments. Application of single treatment or its combined use promoted plant growth and development which is a sign that soil is in good health. Plant spike length and height of rice plants increased significantly after the inoculation of AMF and PGPR. Hence, it is clear from the experiments conducted that local application of AMF and PGPRs improved nutrient contents in soil and significantly enhanced their uptake in rice plants. This type of treatment can be very much effective for soil reclamation and plant yields for future studies and implementation for the stakeholders”.
